# Coronavirus Pandemic—Therapy and Vaccines

**DOI:** 10.3390/biomedicines8050109

**Published:** 2020-05-03

**Authors:** Kenneth Lundstrom

**Affiliations:** PanTherapeutics, CH1095 Lutry, Switzerland; lundstromkenneth@gmail.com; Tel.: +41-79-776-6351

**Keywords:** coronavirus, peptide vaccine, viral vectors, gene silencing, RNA interference, viral replication

## Abstract

The current coronavirus COVID-19 pandemic, which originated in Wuhan, China, has raised significant social, psychological and economic concerns in addition to direct medical issues. The rapid spread of severe acute respiratory syndrome-coronavirus (SARS-CoV)-2 to almost every country on the globe and the failure to contain the infections have contributed to fear and panic worldwide. The lack of available and efficient antiviral drugs or vaccines has further worsened the situation. For these reasons, it cannot be overstated that an accelerated effort for the development of novel drugs and vaccines is needed. In this context, novel approaches in both gene therapy and vaccine development are essential. Previous experience from SARS- and MERS-coronavirus vaccine and drug development projects have targeted glycoprotein epitopes, monoclonal antibodies, angiotensin receptor blockers and gene silencing technologies, which may be useful for COVID-19 too. Moreover, existing antivirals used for other types of viral infections have been considered as urgent action is necessary. This review aims at providing a background of coronavirus genetics and biology, examples of therapeutic and vaccine strategies taken and potential innovative novel approaches in progress.

## 1. Introduction

Coronaviruses belong to the family *Coronaviridae* and are characterized by causing respiratory tract infections ranging from mild diseases such as common cold to pneumonia with a lethal outcome [1]. Typically, coronaviruses have been associated with a large number of diseases in livestock and companion animals such as pigs, cows, chickens, cats and dogs (Table 1) [2]. In this context, transmissible gastroenteritis virus (TGEV) [3] and porcine epidemic diarrhea virus (PEDV) [4] are responsible for significant morbidity and mortality in young piglets. Similarly, porcine hemagglutinating encephalomyelitis virus (PHEV) causes enteric infection in pigs but can also lead to encephalitis by targeting the nervous system [5]. In cattle, bovine CoV (BCoV) is responsible for mild to severe respiratory tract infections, resulting in significant losses in the cattle industry due to diarrhea, dehydration, decreased milk production and depression [6,7]. In addition to cattle, BCoV also infects other ruminants such as elk, deer and camels. Another coronavirus, rat CoV (RCoV), causes respiratory tract infections in rats, providing a useful model for studying early events of innate immune responses to coronavirus infections in lungs [8]. Infectious bronchitis virus (IBV) targets chickens, causing respiratory tract infections but also renal disease [9]. IBV has a significant negative effect on egg production and growth of chickens, leading to substantial losses in the chicken industry [7]. In domestic cats, a mild or asymptomatic infection has been associated with feline enteric coronavirus (FCoV) [10], although a highly virulent strain of feline infectious peritonitis virus (FIPV) causes lethal feline infectious peritonitis (FIP) [11], which shows similarities to human sarcoidosis [12].

Interestingly, wildlife can also be infected by coronaviruses. For instance, a novel coronavirus SW1 was isolated from a deceased Beluga whale [13]. The whale had suffered from a respiratory disease and acute liver failure and large quantities of SW1 particles were isolated from the liver. However, electron microscopy was not able to identify the virus as a coronavirus, but nucleic acid sequencing of liver tissue confirmed the presence of coronavirus RNA. Phylogenetic analysis indicated that SW1 belonged to the group of γ-coronaviruses. For obvious reasons, there has been an intense interest in bat coronaviruses, as bats have been indicated as the original source of several coronavirus outbreaks. In the past decade, hundreds of novel bat coronaviruses have been identified [14]. In this context, both the severe acute respiratory syndrome-coronavirus (SARS-CoV) and the Middle East respiratory syndrome-CoV (MERS-CoV) have been suggested to originate from bats. Moreover, it has been already discussed in 2015 that a SARS-like cluster of bat coronaviruses could pose a threat for human epidemics [15]. For instance, the disease potential of a SARS-like virus, SHC014-CoV, circulating in Chinese horseshoe bats was evaluated by reverse genetics [15]. The chimeric virus composed of SHC014-CoV spikes and wild-type backbone was able to efficiently utilize multiple orthologs of the SARS receptor, human angiotensin-converting enzyme 2 (ACE2), replicate to high levels in primary human airway cells and generate in vitro titers equivalent to epidemic SARS-CoV strains. Moreover, the chimeric virus showed replication in mouse lung, with notable pathogenesis in vivo. These results indicated that there is a potential risk of SARS-CoV re-emergence from viruses circulating in bat populations. Needless to say, the re-emergence of SARS-CoV is a reality today, as will be discussed below.

The murine hepatitis virus (MHV) is the best studied animal coronavirus resulting in respiratory, enteric, hepatic and neurological infections and is also a useful disease model [16]. For instance, MHV-1 causes severe respiratory infections in A/J and C3H/HeI mice. A59 and MHV-3 are associated with severe hepatitis, and JHMV causes severe encephalitis. Interestingly, A59 and an attenuated JHMV cause chronic demyelinating disease in mice, which resembles multiple sclerosis (MS), providing a mouse model for MS [17]. Another advantage of MHV is the requirement of BSL2-level laboratory conditions, whereas SARS-CoV and MERS-CoV require BSL3-level laboratory conditions, making MHV ideal for studies on replication in cell cultures as well as pathogenesis and immune responses in vivo.

In the context of human coronaviruses, it was thought that they caused only mild self-limiting infections until the SARS-CoV outbreak in 2002–2003 [2]. Two human α-coronaviruses (HCoV-229E and HCoV-NL63) and two β-coronaviruses (HCoV-OC43 and HCoV-HKU1) were identified as endemic in human populations, responsible for 15%–30% of annual respiratory tract infections [18,20]. However, a more severe disease has been detected in neonates, elderly people and in individuals with pre-existing illnesses. Moreover, HCoV-NL63 has also been associated with acute laryngotracheitis [19]. Interestingly, human coronaviruses differ in genetic variability, as HCoV-229E isolated from around the world showed only minimal sequence divergence [23,24,25,26,27,28,29,30], while HCoV-OC43 isolated from the same location in different years demonstrated significant genetic variability [31]. For this reason, HCoV-229E does not cross species barriers and is incapable of infecting mice, while mice and several ruminant species are susceptible to HCoV-OC43 and the closely related BCoV. As mentioned above, although MHV possesses the ability to cause demyelinating disease, no evidence exists of human coronaviruses being associated with MS.

The first major human coronavirus outbreak caused by the β-coronavirus SARS-CoV originated in Guangdong in China [21]. During the 2002–2003 epidemic, approximately 8098 cases were recorded, with 774 deaths at a mortality rate of 9% [22]. However, the mortality rate was much higher in the elderly population, reaching almost 50% in individuals over 60 years of age. Although closely related viruses were isolated from exotic animals such as Himalayan palm civets and raccoon dogs [24], based on sequence data and serologic evidence, SARS-CoV originated in Chinese horseshoe bats [25,26]. Related to SARS-CoV, two novel bat SARS-related CoVs with closer similarity to SARS-CoV than any other virus, were identified in 2013 [27]. They utilize the same receptor, ACE2, as SARS-CoV, another indication that SARS-CoV originated in bats. The SARS outbreak was only spread through direct contact with infected individuals due to the relatively inefficient transmission of the virus, which allowed containing the outbreak within households and health care institutions [28]. Therefore, the SARS-CoV outbreak was controllable through quarantining, and it died out in June 2003.

After the SARS-CoV epidemic, the novel human MERS-CoV emerged in the Middle East in 2012, causing a series of highly pathogenic respiratory tract infections in Saudi Arabia and other Middle East countries [29]. Despite fears, the outbreak did not accelerate in 2013 and the cases of MERS-CoV totaled at 855 individuals and 333 deaths, resulting in a mortality rate of almost 40% [23]. MERS-CoV is a β-coronavirus, which is highly related to the bat coronaviruses HKU4 and HKU5 [32], suggesting its origin in bats, although dromedary camels have been indicated as an intermediate host [33]. Moreover, it was demonstrated that MERS-CoV replicated in camel cell lines, further indicating that camels may be natural hosts for the virus [34]. Additionally, a case study showed that virus isolated from a person who had been in contact with an infected camel was identical to virus isolated from the camel [35]. In contrast to SARS-CoV, the dipeptidyl peptidase 4 (DPP4) receptor is utilized, making bats, humans, camels, rabbits, and horses susceptible [36]. However, differences in the structure of the mouse DPP4 receptor make mouse cells resistant to MERS-CoV, which does not allow evaluation of antivirals and vaccines in appropriate mouse models. However, a mouse model for MERS-CoV was engineered by introduction of the human DPP4 gene into mouse lungs by an adenoviral vector [37].

The current 2019-nCoV, officially called severe acute respiratory virus-coronavirus-2 (SARS-CoV-2) was first detected in the city of Wuhan in China in December 2019 [38,39]. It was thought to have originated from wild animals at the Huanan market in Wuhan and spread by person-to-person transmission, causing the disease named COVID-19, with various degrees of severity, from mild flu-like symptoms to pneumonia and death [39]. Bats, snakes and pangolins are potential carriers of SARS-CoV-2 based on sequence comparison to virus isolated from SARS-CoV-2-infected patients. The COVID-19 pandemic has already overtaken both SARS and MERS in severity and has recently led to extreme quarantine measures including sealing off large cities, closing borders and confining people to their homes. Despite these efforts, as of April 28, there are 3,041,912 cases of COVID-19 and there have been 211,167 deaths in 210 countries [40]. Whether the trend will be a further expansion of the pandemic and when it will die out or return in a seasonal pattern is impossible to determine at this stage and only time will tell the outcome.

Currently, there are no antiviral drugs or vaccines available for the treatment of COVID-19 [41]. The urgent need for treatment has therefore triggered a variety of approaches to design and develop novel drug and vaccine strategies against COVID-19. Therapeutic approaches have included angiotensin receptor blockers, monoclonal antibodies (mAbs), gene silencing and even plant and mushroom extracts based on traditional Chinese medicine. Moreover, vaccine development involves different alternatives including epitope-based peptide vaccines, viral vector-based vaccines and the support of bioinformatics and immunoinformatics for the design of more efficient targets. However, before the various approaches are described in detail, it is appropriate to give an overview of the genomic composition of SARS-CoV-2 and its lifecycle to identify potential drug and vaccine targets.

## 2. Coronavirus Genome and Lifecycle

Coronaviruses possess a non-segmented positive-sense single-stranded RNA (ssRNA) genome of approximately 30 kb [42,43]. It contains a 5′ end cap structure, replicase genes (rep 1a and rep 1b) coding for the non-structural proteins (nsPs) comprising two-thirds of the genome and the structural protein genes S (spike), E (envelope), M (membrane) and N (nucleocapsid) and various accessory genes interspersed within the structural genes at the 3′ end [2] (Figure 1). Although the accessory proteins have been considered non-essential for replication, they play an important role in viral pathogenesis [43].

The virion structure has been studied by cryo-electron tomography and microscopy, showing the prominent club-shaped spike projections on spherical particles of approximately 125 nm in diameter [44,45]. The N protein is helically symmetrical, with both N- and C-terminal domains needed for optimal encapsulation of the RNA genome [46]. The transcription regulation sequences (TRSs) [47] and the genomic packaging signal [48] have been identified—of which, the latter has been demonstrated to bind specifically to the C-terminal RNA-binding domain facilitating the packaging of the genome into viral particles [49]. The M protein containing three transmembrane domains may exist as a dimer, binds to the N protein [50] and is thought to give the virion its shape [51]. The topology of the small 8–12 kD E protein is not completely resolved but is believed to have a transmembrane structure with an N-terminal ectodomain and a C-terminal endodomain with ion channel activity [52]. It facilitates the assembly and release of virions and has been associated with pathogenesis [53]. The spikes consisting of the S protein give the virion the appearance of a solar corona and mediate the attachment to the host receptor [54]. The S protein is cleaved by a host cell furin-like protease into S1 and S2 polypeptides for most coronaviruses [55,56]. However, SARS-CoV and SARS-related CoV lack the furin cleavage site, while SARS-CoV-2 accommodates it [57]. A fifth structural protein, the hemagglutinin-esterase (HE), is present in the genome of some coronaviruses such as MHV [58]. The function of the HE is thought to be the enhancement of the S protein-mediated cell entry and spread through the mucosa [59] as well as increase of the neurovirulence of MHV [60].

The coronavirus lifecycle is characterized by various steps including attachment and entry, expression of replicase proteins, replication and transcription, and assembly and release of mature viral particles (Figure 2). The initial attachment occurs between the receptor-binding domain (RBD) of the S1 region of the S protein and its receptor. MHV uses an RBD at the N-terminus [61], while SARS-CoV has an RBD at the C-terminus of the S1 region [62]. The S protein–receptor interaction governs the virus tropism, which also reflects the targeting of different receptors by different coronaviruses. For instance, MHV targets carcinoembryonic antigen-related adhesion molecule 1 (CEACAM1) [63], MERS-CoV dipeptidyl-peptidase 4 (DPP4) [64] and SARS-CoV and HCoVNL63 ACE2 [65]. Recently, it was demonstrated that SARS-CoV-2 also uses ACE2 [21]. After the initial attachment, acid-dependent proteolytic cleavage of the S protein by cathepsin or another protease occurs, followed by fusion in acidified endosomes or at the plasma membrane, which leads to the release of the viral genome into the cytoplasm [66,67].

The translation of the coronavirus replicase genes from two large open reading frames (ORFs), rep1a and rep1b, allows the expression of two polyproteins, pp1a and pp1ab, utilizing a slippery sequence (5′-UUUAAAC-3′) and an RNA pseudoknot [68,69]. The complicated composition of the pp1a and pp1ab polyproteins and the assembly of nsPs are described in detail by Fehr and Perlman [2]. Following the translation and assembly of the replicase complex, viral RNA synthesis from both genomic and subgenomic RNAs occurs. The latter serves as mRNAs for the structural and accessory genes [2]. In the 5′ untranslated region, there are cis-acting sequences such as stem–loop structures, which are important for viral RNA replication [70,71,72,73]. An interesting feature of coronavirus replication relates to the fusing of the leader and body TRS segments during the production of subgenomic RNAs. Although originally thought to occur during positive-strand synthesis, a recent model suggests that RNA-dependent RNA polymerase (RdRp) pauses at any of the body TRS sequences during the discontinuous extension of negative-strand RNA, followed by either continued elongation to the next TRS or switching to amplification of the leader sequence at the 5′ end [74,75].

In the context of assembly, the viral structural proteins S, E, and M are translated and transported along secretory pathways into the endoplasmic reticulum (ER) and the ER–Golgi intermediate compartment (ERGIC) [76,77]. At the ERGIC, viral genomes surrounded by N protein participate in the formation of mature virions [78]. Assembled virions are then transported to the cell surface in vesicles and released by exocytosis [79]. An interesting observation relates to coronaviruses assembled without the S protein, which leads to the formation of giant multi-nucleated cells spreading the virus within an infected organism without being detected or neutralized by virus-specific antibodies [2].

## 3. Therapeutic and Prophylactic Options

Since the SARS-CoV outbreak in 2002–2003, numerous approaches have been used to develop therapeutic agents and vaccines for coronaviruses (Table 2). The recent COVID-19 pandemic has further accelerated the efforts to find cure and protection. Typically, three general methods are employed for antiviral treatment of coronaviruses [80]: (i) application of existing broad-spectrum antiviral drugs; (ii) screening of chemical libraries containing many existing compounds or databases; (iii) redevelopment of new specific drugs based on the genome and biophysical understanding of individual coronaviruses. Other options include monoclonal antibodies and viral receptor blockers [81]. Products developed as Chinese traditional medicines have also been suggested [80]. In line with gene therapy applications, the design of various gene silencing approaches has been initiated based on RNA interference (RNAi) [82]. In the context of vaccine development, a wide variety of immunization strategies have been initiated [83].

### 3.1. Therapeutic Agents

Among therapeutics, the use of protease inhibitors lopinavir and ritonavir as a combination therapy with other antiretroviral agents for the treatment of HIV-1 infections has provided durable virologic suppression and improved immunological outcomes [84]. Lopinavir/ritonavir combination treatment was evaluated in marmosets with a MERS-like disease [85]. In comparison to mycophenolate mofetil (MFF), the treatment showed improved clinical, radiological and pathological outcomes and lower mean viral loads in necropsied lung and extrapulmonary tissues, whereas MFF-treated animals developed severe or fatal disease. Based on these results, a study protocol for a clinical trial for hospitalized adult MERS patients was prepared [86]. The goal is to investigate laboratory-confirmed MERS patients in a recursive, two-stage, multi-center, placebo-controlled, double-blind randomized controlled trial. Related to the recent COVID-19 pandemic, a clinical trial on 99 patients receiving lopinavir and ritonavir were compared to 100 patients subjected to standard care. However, no difference in clinical improvement, mortality or detectable viral RNA was obtained [87]. In a modified intention-to-treat analysis, the median time to clinical improvement was one day shorter for lopinavir/ritonavir treatment and although gastrointestinal adverse events were more common, serious adverse events were less frequent. Overall, the study indicated that the treatment offered no benefit compared to standard care. Another approach comprises the application of nucleoside analogues as antivirals [88]. For instance, several classes of nucleoside analogues were verified against SARS-CoV in Vero cells [89]. In the study, the D-isomer of thymine analogue exhibited strong anti-SARS-CoV activity and did not show any toxicity at the highest tested dosage of 100 µM. Although the L-cytosine analogue showed good activity, it exhibited strong toxicity to cells. Likewise, the 3′-azido-2′, 3′-unsaturated thymine analogue provided strong anti-SARS-CoV activity, but also significant toxicity. Additionally, dioxalane-thymine showed moderate antiviral activity without any significant cytotoxicity. Although C-nucleoside,4-amino-7-(β-L-ribofuranosylpyrrolo [3,2-D] pyrimidine hydrochloride inhibited SARS-CoV replication, it was cytotoxic. Overall, despite several classes of nucleoside analogues against SARS-CoV exhibiting moderate antiviral activity in vitro, no clear structure–activity relationship could be established. In another study, a series of doubly flexible nucleoside analogues were designed based on the acrylic sugar scaffold of acyclovir [90]. One compound displayed selective antiviral activity against HCoV-NL63 and MERS-CoV. In contrast, no activity was detected against SARS-CoV. In a study on FIP, the nucleoside analogue GS-441524, a precursor to the pharmacologically active nucleoside triphosphate molecule, and acting as an alternative substrate and RNA chain terminator of viral RdRp was non-toxic and effectively inhibited FIPV replication in feline CRFK cells [91]. Moreover, all ten cats treated with GS-441524 showed rapid reversal of disease symptoms in two weeks. In another approach, the guanosine analogue ribavirin was administered together with interferon-α2a in adult patients with laboratory-confirmed MERS-CoV infection and pneumonia needing ventilation support [92]. The study demonstrated that patients treated with ribavirin and interferon-α2a showed a significantly improved survival rate at 14 days but not at 28 days of treatment compared to the control group of patients. The adenosine nucleoside analogue remdesivir has proven efficient for inhibition of RNA viruses such as filoviruses, pneumoviruses and paramyxoviruses by targeting RdRp [93]. Moreover, the antiviral activity of remdesivir has also been demonstrated for human endemic and zoonotic coronaviruses such as HCoV-OC43 and HCoV-229E [94]. In another study, it was shown that remdesivir potently inhibited RdRp from MERS-CoV [95]. Very recently, it was demonstrated that remdesivir effectively inhibited SARS-CoV-2 in Vero cells and in human Huh-7 liver cancer cells known to be susceptible to SARS-CoV-2 [96]. The first case of compassionate treatment of COVID-19 with intravenous administration of remdesivir took place in the US, leading to improvement in the patient’s condition with a decline in viral load [97]. Neuraminidase inhibitors such as zanamivir, laninamivir, oseltamivir and peramivir have demonstrated potency against most influenza strains [98]. In this context, the neuraminidase inhibitors oseltamivir (Tamiflu) and zanamivir (Relenza) were tested for inhibition of cytopathic effects of SARS-CoV in a cytopathic endpoint assay in Vero cells [99]. However, neither neuraminidase inhibitor showed any effect on SARS-CoV. Arbidol, a broad-spectrum antiviral agent against several DNA and RNA viruses, has been approved in Russia and China for prevention and treatment of influenza [100]. Antiviral effect against Zika virus (ZIKV), West Nile virus (WNV) and tick-borne encephalitis virus (TBEV) has been demonstrated in Vero cells [100]. Furthermore, it was shown that arbidol can inhibit six different isolates of ZIKV and can protect against the cytopathic effects of ZIKV [101]. Related to SARS-CoV, patients with laboratory-confirmed COVID-19 were treated with oral arbidol in combination with lopinavir/ritonavir or lopinavir/ritonavir monotherapy [102]. After seven days, SARS-CoV could not be detected in nasopharyngeal specimens in 12 out of 16 patients subjected to the combination therapy compared to 6 out of 17 for the monotherapy group. The numbers at day 14 were 15 out of 16 for combination therapy and 9 out of 17 for monotherapy at day 14. Furthermore, CT scans revealed improvement for 11 out of 16 patients receiving the combination therapy, and 5 out of 17 receiving the monotherapy.

**Table 2 biomedicines-08-00109-t002:** Therapeutic interventions against coronaviruses.

Therapy	Disease	Effect	Ref.
Lopinavir/Ritonavir	MERS-like in marmosets	Clinical, pathological benefits	[85]
	MERS	Protocol for clinical trial	[86]
	COVID-19	No difference to standard care	[87]
Nucleoside	SARS-CoV	In vitro anti-SARS-CoV activity	[89]
analogues	HoCV-NL63, MERS-CoV	Antiviral activity, but not against SARS-CoV	[90]
	FIP	Successful treatment of FIP	[91]
Ribavirin	MERS-CoV	Improved survival at 14 days	[92]
Remdesivir	HCoV-OC43, HCoV-229E	Potent antiviral activity	[94]
	MERS-CoV	Potent inhibition	[95]
	SARS-CoV-2	Inhibition in Vero, Huh-7 cells	[96]
	SARS-CoV-2	Reduced viral load in patients	[97]
NA inhibitors	SARS-CoV	No inhibitory effect	[99]
Arbidol	SARS-CoV-2	Positive effect in patients with lopinavir/ritonavir	[102]
Hydroxychloroquinine	SARS-CoV-2	Reduced viral load in 20 COVID-19 patients	[103]
Chinese medicines	SARS-CoV-2	Case studies of recovery from COVID-19 by SHL	[104]
	SARS-CoV-2	Successful SJ therapy	[105]
ACE2	SARS-CoV-2	nAbs for compassionate therapy	[81]
	SARS-CoV-2	TMPRSS2 blocking virus entry	[106]
	SARS-CoV-2	Potent binding of mAbs	[107]
ATR1 blockers	SARS-CoV-2	Evaluation of existing blockers	[108]
mAb	SARS-CoV, SARS-CoV-2	Neutralization of SARS-CoV-2	[109]

ACE2, angiotensin-converting enzyme 2; CoV, coronavirus; FIP, feline infectious peritonitis; HCoV, human coronavirus; MERS, Middle East respiratory syndrome; NA, neuraminidase; nABs, neutralizing antibodies; mAbs, monoclonal antibodies; SARS-CoV, severe acute respiratory syndrome-coronavirus; SHL, Shuanghuanglian oral liquid; SJ, Shufeng Jiedu; TMPRSS2, transmembrane protease, serine 2.

A hot topic today has been the potential therapeutic effect of the widely used antimalarial and auto-immune disease drug chloroquine on coronaviruses [96]. Chloroquine blocks virus infection and interferes with the glycosylation of cellular receptors of SARS-CoV [110]. The application of chloroquine for COVID-19 patients has been justified by its common use by travelers in malaria-endemic geographic regions for decades and continuous use by locals [111]. However, as indications of severe side effects of chloroquine use have been reported, hydroxychloroquine, possessing a similar antiviral potency to chloroquine but a safer clinical profile, should be considered [112]. In a limited clinical trial, 20 French COVID-19 patients were treated with hydroxychloroquine, resulting in a significant reduction in viral load in comparison to control patients [103]. Addition of azithromycin further significantly improved the efficiency of virus elimination. Despite the small sample size, the study indicated that hydroxychloroquine treatment was associated with a significant viral load reduction and disappearance of COVID-19. However, the trial design was poor and the results unreliable, as six patients dropped out and the assessment of efficacy was based on viral load, which was not a clinical endpoint. Additional studies are therefore needed in a larger number of patients to thoroughly validate the safety and efficacy of hydroxychloroquine.

Chinese traditional medicine has also tried to provide support in the fight against coronaviruses. In this context, case studies of treatment with the Shuanghuanglian oral liquid (SHL) of a family suffering from COVID-19 has been reported [104]. Case 1, a 51-year-old female with high fever diagnosed with COVID-19 was confined into an isolation ward five days later, where she started to take oral SHL twice a day. The following day, SHL administration was increased to three doses a day without taking any other drugs. The fever decreased in two days, and she gradually recovered. Case 2, the 27-year-old daughter of Case 1 presented with a high fever, vomiting and diarrhea. Two days after she was confined in an isolation ward, she started to take SHL three times a day. A couple of days later, a decrease in body temperature and recovered appetite were noted. Case 3, the husband of Case 1, had mild diarrhea, vomiting and fever and was diagnosed with COVID-19. He started to take SHL three times a day, as well as moxifloxacin and arbidol, which resulted in all symptoms with exception of light nausea disappearing within four days. In another study, four patients with mild or severe 2019-nCoV pneumonia were treated with lopinavir/ritonavir, arbidol and traditional Chinese medicine in the form of Shufeng Jiedu capsules [105]. Three patients showed significant improvement related to the pneumonia symptoms and tested 2019-nCoV negative, and one patient with severe pneumonia also showed improvement. Another potential approach for COVID-19 treatment relates to AHCC, an α-glucan-based standardized mushroom extract from *Lenintula edodes*, which has demonstrated immunostimulation in humans infected by WNV, influenza virus, avian influenza virus, hepatitis C virus, papilloma virus, herpes virus, hepatitis B virus and HIV [113]. Although the potency of AHCC has not yet been verified for SARS-CoV-2, it might be an attractive alternative approach to explore. Likewise, lianhuaqingwen capsules have proven to be efficient for the prevention and treatment of respiratory infections caused by influenza A [114] but have not yet been validated for SARS-CoV-2.

In light of therapeutic strategies to rapidly target SARS-CoV-2, one approach aims at blocking virus entry using a soluble version of ACE2 fused to an immunoglobulin Fc domain [81]. This approach will elicit broad-ranging neutralizing antibodies and stimulate the immune system. Recombinant protein expression can be utilized for the rapid production of a drug for compassionate use while formal clinical trials are undertaken, and vaccines developed. Another target related to ACE2 comprises the transmembrane protease serine 2 (TMPRSS2) involved in S protein priming, for which TMPRSS2 inhibitors might be approved for clinical use to block coronavirus entry [106]. It was also demonstrated that sera from convalescent SARS patients are capable of cross-neutralizing SARS-CoV-2 S-driven entry [106]. In the context of ACE2, although the binding affinity of SARS-CoV and SARS-CoV-2 is similar, the presence of the furin cleavage site uniquely in the SARS-CoV-2 S protein sets it apart for the design of specific inhibitors [57]. Moreover, the finding that SARS-CoV S murine polyclonal antibodies can potently prevent SARS-CoV-2 S-mediated cell entry indicated that cross-neutralizing antibodies against conserved S epitopes might be a feasible approach for therapy. In the context of therapeutic monoclonal antibodies for SARS-CoV-2, the first SARS-CoV-specific human monoclonal antibody (mAb) CR3022 showed potent binding to the SARS-CoV-2 receptor-binding domain [107]. However, the CR3022 epitope does not overlap with the ACE2 binding site, which indicates that CR3022 could have the potential as a therapeutic as such or in combination with other neutralizing antibodies. Interestingly, the potent SARS-CoV-specific m396 and CR3014 neutralizing antibodies, which target the ACE2 binding site, failed to bind the SARS-CoV-2 S protein, suggesting differences in the receptor-binding domains of SARS-CoV and SARS-CoV-2. It is therefore necessary to develop mAbs with specific binding affinity to the SARS-CoV-2 receptor-binding domain. Another tentative approach comprises the evaluation of existing angiotensin receptor 1 (ATR1) blockers such as losartan, commonly used for the treatment of hypertension [108], for a reduction in aggressiveness and mortality from SARS-CoV-2 infections [115]. The approach is based on the finding that ACE2 most likely represents the binding site for both SARS-CoV and SARS-CoV-2, thereby providing a sensible target for therapeutic interventions for coronavirus infections. Targeting ATR1 might also address the problem with new emerging coronavirus mutations. Very recently, the first report on the human 47D11 mAb targeting a conserved epitope in the spike receptor-binding domain was published [109]. It cross-neutralizes SARS-CoV and SARS-CoV-2 independently from receptor-binding inhibition and will be useful for the development of antigen detection tests and serological assays. Moreover, it adds to the potential to prevent and treat COVID-19 and possible future emerging coronaviruses.

### 3.2. Gene Silencing

Gene silencing based on RNA interference has proven to be an important tool in basic research but also for therapeutic applications [116,117]. The mechanism of RNAi relates to 19–23 base pair double-stranded RNAs (dsRNAs) mediating degradation of target RNA in a sequence-specific manner [118]. Coronaviruses have been subjected to several gene silencing studies (Table 3). For instance, short interfering RNAs (siRNAs) have been demonstrated to efficiently inhibit SARS-CoV replication in Vero E6 cells [119]. Moreover, 48 siRNA sequences were designed throughout the SARS-CoV genome, targeting several key proteins [120]. Chemically synthesized siRNAs were transfected into fetal rhesus kidney FRhK4 cells before or after SARS-CoV infection and the inhibitory effects were verified by the decrease in intracellular viral genome copy number and viral titers. Four siRNAs demonstrated potent inhibition of SARS-CoV infection and replication. Prophylactic effects with up to 90% inhibition lasted at least for 72 h. Combination of siRNA duplexes from different regions of the viral genome provided up to 80% inhibition. Furthermore, siRNA duplexes have been shown to significantly suppress SARS-like symptoms in vivo in rhesus macaques [121]. In another approach, expression of U6 promotor-driven siRNA homologous to ACE2 mRNA silenced ACE2 expression in Vero cells [122]. It was further demonstrated that SARS-CoV infection was reduced in ACE2-silenced cell lines, providing an attractive approach for siRNA-based prophylactic or therapeutic strategies. Moreover, siRNA duplexes were applied to knock down expression of the actin-binding protein ezrin, which interacts with the SARS-CoV spike protein during the entry stage of infection [123].

In the context of MERS-CoV the ORF1ab region encoding the replicase polyproteins plays a vital role in viral infection and therefore represents a suitable target for disease control. Four miRNA and five siRNA molecules were rationally designed by computational methods for silencing of nine different MERS-CoV strains for exploration of the treatment of MERS-CoV at the genomic level [124]. In attempts to optimize delivery of MERS-CoV siRNAs, transfection, electroporation and viral gene transfers have been applied. Recently, advanced nanotechnology based on lipids, polymers and inorganic compounds have been formulated [128]. Gene silencing of FIPV by siRNAs has been hampered by mutations creating resistant viruses. To address the problem, combination therapy with three siRNAs prevented viral escape over the course of five passages [125]. Moreover, Dicer-substrate siRNAs provided equivalent or better potency than canonical siRNAs for FIPV. In the context of porcine deltacoronavirus (PDCoV), two short hairpin RNA (shRNA)-expressing plasmids targeting the M (pGenesil-M) and N (pGenesil-N) genes of PDCoV were evaluated in swine testicular (ST) cells [126]. Challenges with the PDCoV HB-BD strain provided highly specific and efficient protection of ST cells. Treatment with pGenesil-M and pGenesil-N resulted in a 13.2- and 32.4-fold titer reduction, respectively, and a 45.8% and 56.1% decrease in viral RNA, respectively. In another study, shRNAs targeting the M gene of porcine epidemic diarrhea virus (PEDV) and swine acute diarrhea syndrome coronavirus (SADS-CoV) and the N gene of PDCoV inhibited expression of each viral RNA over 98% [127]. Moreover, shRNAs significantly restricted the expression of M and N proteins and impaired PEDV, SADS-CoV and PDCoV replication.

### 3.3. Vaccine Development

The classic approach for vaccine development against viral diseases has involved immunization with live attenuated or inactivated viruses [129]. The availability of genetic engineering and efficient recombinant protein production technologies has shifted applications to the utilization of recombinantly expressed antigens and immunogens for immunization [130]. In the context of coronaviruses, vaccine development started seriously after the SARS and MERS outbreaks, providing alternative approaches of applying subunit vaccines, whole inactivated virus, vectored, and live attenuated virus vaccines [83] (Table 4). Efforts to tackle other coronaviruses have also been explored. For instance, a modified-live vaccine against BCoV was developed by progressive attenuation of the respiratory BCoV strain 438/06-TN [131]. The vaccine was proven safe and intramuscular injection in calves elicited high antibody titers against BCoV 30 days post-vaccination. The intranasal drug Bovilis^®^ has been approved against enteric disease caused by BCoV in young calves [132]. The lack of relevant vaccines for equine coronavirus (ECoV) catalyzed the study on antibody responses to ECoV in horses after vaccination with the BCoV vaccine [133]. Antibody titers against ECoV increased in all six vaccinated horses at 14 days post-inoculation, although the titers were lower against ECoV than BCoV, and it remained unclear whether the elicited antibodies provided protection against ECoV.

In the context of vaccine development, different computational and informatics tools play an essential role. For instance, the immune epitope database (IEDB) has been used to predict suitable MERS-CoV epitope vaccines against the most known world population alleles based on the S and E proteins [134]. The study showed that highly conserved sequences in the S and E proteins might be considered immunogenically protective and can elicit both neutralizing antibodies and T cell responses when reacting with B cells, T helper cell lymphocytes (HTLs) and cytotoxic T lymphocytes (CTLs). In another approach, the SARS-CoV-2 S protein was characterized to obtain immunogenic epitopes for vaccine development [135]. Thirteen major histocompatibility complex (MHC)-I and three MHC-II epitopes with antigenic properties were identified. The epitopes were linked by specific linkers and docked to toll-like receptor-5 (TLR5), and immunoinformatics analysis was utilized for fast immunogenic profiling to accelerate vaccine development. In another immunoinformatics and computational approach, conserved B and T cell epitopes for the MERS-CoV S protein were identified [136]. The antigenicity of the epitopes and interactions with the human leukocyte antigen (HLA) B7 allele were estimated. The highest antigenicity score was obtained for the immensely immunogenic B cell epitope QLQMGFGITVQYGT. The T cell epitope peptides YKLQPLTFL (MHC-I) and YCILEPRSG (MHC-II) were also highly antigenic. These identified putative antigenic epitopes may prove effective for the development of novel vaccines. Using in silico approaches, two multi-epitope vaccines against MERS-CoV were designed by screening CTL and HTL epitopes from 13 different MERS-CoV proteins [137]. Both multi-epitope vaccines also carried potential B cell linear epitope regions, B cell discontinuous epitopes and interferon-γ-inducing epitopes. Moreover, human β-defensin-2 and β-defensin-3 were used as adjuvants for enhanced immune responses. The most potent CTL and HTL epitopes and adjuvants were linked by short peptide molecular linkers. Tertiary models for both multi-epitope vaccines were verified for their molecular interaction with TLR3 and cDNAs were generated for in silico analysis of expression in human cell lines before being tested in vivo as potential vaccine candidates.

Computational approaches and immunoinformatics have provided strong support for vaccine development as described above. Moreover, structure–function studies have also contributed to the field. In this context, in comparison to antibodies targeting the receptor-binding domain on the MERS-CoV S protein, less attention has been paid to antibodies targeting non-receptor-binding domain epitopes such as the neutralizing antibody G2, which targets the MERS-CoV N-terminal domain of S1 [138]. Structural and functional characterization of G2 alone or complexed with the MERS-CoV N-terminal of S1 demonstrated that G2 strongly disrupts the attachment of MERS-CoV S to the DDP4 receptor and could play an important role as a target for immunotherapy and vaccine development. In another approach, cryo-electron microscopy (cryo-EM) was applied for the human HCoV-NL63, providing a 3.4 Å resolution of the spike glycoprotein trimer essential for viral entry into host cells and representing the main target for neutralizing antibodies [139]. The structure revealed important components of the fusion process including the triggering loop and the C-terminal domains involved in the anchoring of the trimer to the viral membrane. The study also revealed that HCoV-NL63 use molecular trickery based on epitope masking with glycans and activating conformational changes in attempts to evade recognition by the host immune system. In another study, the SARS-CoV-2 S protein was subjected to cryo-EM structure determination of the ectodomain trimer for the identification of potential targets for vaccines and viral entry inhibition as the furin cleavage site between the S1 and S1 subunits is unique for SARS-CoV-2 and not present in SARS-CoV or other related coronaviruses [57]. In a study on SARS-CoV, screening of experimentally determined B and T cell-derived epitopes from the S and N proteins, which maps identically to SARS-CoV-2, showed no mutations among the 120 available SARS-CoV-2 sequences [140]. Immune targeting of these epitopes may therefore potentially offer protection against SARS-CoV-2. Related to the T cell epitopes, a population coverage analysis of associated MHC alleles was conducted, allowing the identification of a set of epitopes, which might provide a broad global coverage.

**Table 4 biomedicines-08-00109-t004:** Vaccine approaches against voronaviruses.

Vaccine/Vector	Disease	Effect	Ref.
Live attenuated	BCoV	High Ab titers against BCoV in calves	[131]
	BCoV	Approved nasal vaccine in calves	[132]
BCoV vaccine	ECoV	Increased Ab titers against ECoV	[133]
Plasmid DNA	SARS-CoV	Humoral and cellular immune responses in mice	[141]
DNA-CTE/PRE	SARS-CoV	nAbs, protection against SARS-CoV in mice	[142]
DNA/PEI NPs	SARS-CoV	NPs induced humoral and cellular responses in mice after intranasal administration	[143]
	TCoV	Humoral response, partial TCoV protection	[144]
DNA + protein	TGEV	Oral administration induced mucosal and cellular immune responses in mice	[145]
*Lactobacillus acidophilus* TGEV	MERS-CoV	nAbs, protection of mice against MERS-CoV	[146]
CHO/S377-588	SARS-CoV	Overexpression of S protein in plants	[147]
Tobacco/lettuce	SARS-CoV	IgA Abs in mice fed with tomato-derived S	[148]
Tomato/tobacco	SARS-CoV	Humoral and cellular immune responses	[149]
Tobacco/suppressor		SARS-CoV N protein	
p19 TBSV	PEDV	Immune response in mice and piglets	[150]
Ad-LTB-COE	MERS-CoV	Reduced viral excretion and viral RNA in dromedary camels	[151]
MVA-MERS-CoV S	MERS-CoV	Identification of T cell-responding epitope	[152]
MVA-MERS-CoV N	SARS-CoV	Strong nAbs response in mice	[153]
RV-SARS-CoV N/S	SARS-CoV	Protection against SARS-CoV in mice	[154]
VEE-SARS-CoV S	SARS-CoV	Protection also in aged mice	[155]

Abs, antibodies; Ad-LTB-COE, adenovirus-based heat-labile enterotoxin B-core neutralizing epitope of PEDV; BCoV, bovine coronavirus; CTE, constitutive transport element from Mason-Pfizer monkey virus; ECoV, equine coronavirus; MERS-CoV, Middle East respiratory syndrome-coronavirus; MVA, Modified vaccinia virus Ankara; P19 TBSV, gene silencing suppressor P19 protein from tomato bushy stunt virus; PEDV, porcine epidemic diarrhea virus; PEI, polyethylenimine; PRE, post-transcriptional regulatory element from Woodchuck hepatitis virus; nABs, neutralizing antibodies; NPs, nanoparticles; S377-588, RV, rabies virus; SARS-CoV, severe acute respiratory syndrome-coronavirus, TCoV, turkey coronavirus; TGEV, transmissible gastroenteritis virus.

Nucleic acid-based vaccines have become attractive alternatives to vaccines based on live attenuated or inactivated viruses. In this context, four plasmid DNA-based vaccine constructs were intradermally administered into C57BL/6 mice [141]. The pLL70 vector contained the SARS-CPoV S gene and the pcDNA-SS vector contained the codon-optimized SARS-CoV S gene fused with the leader sequence from the human CD5 gene. The pcDNA-St vector carried the N-portion of the codon-optimized S gene with the CD5 leader sequence and the pcDNA-St-VP22C contained the N-portion of the codon-optimized S with the CD5 leader sequence fused to the C-terminal of the bovine herpesvirus-1 (BHV-1) VP22 protein, known to facilitate and enhance protein delivery [156]. Immunization studies revealed that pcDNA-SS and pcDNA-St-VP22C elicited superior cellular and humoral immune responses in mice and therefore represent the most immunogenic SARS vaccine candidates. Moreover, the DNA vaccine approach was verified targeting the SARS-CoV S protein using an improved plasmid DNA vector containing donor and acceptor splice sites and heterologous viral RNA export elements such as the constitutive transport element (CTE) and the post-transcriptional regulatory element (PRE) from Mason-Pfizer monkey virus and Woodchuck hepatitis virus, respectively [142]. These vector modifications significantly improved the immunogenicity and immunization of mice with 2 µg of naked DNA induced neutralizing anti-S antibodies and provided protection against challenges with SARS-CoV.

In another plasmid DNA-based study, immune responses to a naked plasmid vector and plasmid/polyethylenimine nanoparticles expressing the SARS-CoV S protein were investigated in BALB/c mice after intranasal administration [143]. Immunization with nanoparticles elicited significantly higher S-specific IgG1 antibodies in sera and mucosal secretory IgA antibodies in lung wash than in mice receiving naked plasmid DNA. In another study, DNA plasmid-based prime immunization was combined with a boost vaccination with a protein [144]. Turkeys were immunized with one or two doses of 750 μg of a DNA plasmid carrying a turkey coronavirus (TCoV) S protein fragment containing neutralizing epitopes (4F/4R) followed by a boost with 200 μg of 4F/4R fragment. Animals were challenged with infectious TCoV and clinical signs were monitored by an immunofluorescence antibody assay. Immunized turkeys showed less clinical signs and a lower viral load compared to control animals. The vaccination also induced humoral immune responses and provided partial protection against challenges with TCoV.

Cellular and viral vector-based expression systems play an important role in vaccine development. The large size of the CoV genome has complicated the construction of infectious clones for utilization in studies on basic viral processes and development of genetically defined vaccines [156]. In this context, bacterial artificial chromosomes (BACs) can provide a robust system for expression of viral RNA in the nucleus under the control of a cytomegalovirus (CMV) promoter followed by RNA amplification by the viral replicase in the cytoplasm [157]. In a prokaryotic approach, a eukaryotic recombinant plasmid expressing the SAD epitope (A and D antigenic sites of the S protein) of TGEV was transformed into *Lactobacillus acidophilus* originating from swine [145]. Oral administration of *L. acidophilus* in BALB/c mice induced significantly higher levels of S IgA antibodies compared to a commercial inactivated TGEV vaccine. The levels of TGEV-specific IgGs were similar but higher levels of interferon-γ were induced by the *L. acidophilus* vaccine. Overall, the oral TGEV *L. acidophilus* vaccine induced high levels of both mucosal and humoral immune responses. In another approach, a stable CHO cell line was engineered to express a subunit recombinant protein vaccine of residues 377-588 of the receptor-binding domain of the MERS-CoV S protein [146]. The vaccine has been demonstrated to elicit significant neutralizing antibody responses and can provide protection against MERS-CoV challenges in vaccinated animals. For stable expression, the IL-2 signal peptide was introduced in front of the S protein domain fused to the human IgG Fc fragment and transfected into an adherent dihydrofolate reductase-deficient CHO cell line. The adCHO-expressed fusion protein was secreted and showed functionality and binding specificity, and a suspension CHO cell line has been developed. Engineered transgenic mice with the DPP4 receptor susceptible to MERS-CoV were immunized with the S377-588-Fc subunit vaccine and adjuvant, which resulted in production of neutralizing antibodies against MERS-CoV and survival for at least 21 days after challenges with live MERS-CoV.

An interesting approach has been to utilize plant expression systems for coronavirus vaccine development. For instance, the N-terminal part (amino acids 1-658) of the SARS-CoV S protein was optimized for codon usage in plants and expressed as a fusion protein with the green fluorescent protein (GFP) in tobacco leaves [147]. It was demonstrated that the S1-GFP fusion protein was expressed in the cytoplasm. Stable expression from the cauliflower mosaic virus 35S promoter resulted in a high level of expression of the fusion protein in tobacco and lettuce leaves. Moreover, S1 production was also achieved in chloroplast-transformed plants, suggesting the potential for developing safe oral plant-derived subunit vaccines against SARS-CoV. In another study, an N-terminal fragment of the SARS-CoV S protein was expressed at high levels in tomato and low-nicotine tobacco plants [148]. The plant-derived antigen elicited systemic and mucosal immune responses in mice. Significantly increased levels of SARS-CoV-specific IgA antibodies were detected in mice after oral administration of tomato fruits expressing the S1 protein. Moreover, SARS-CoV-specific IgG antibodies were detected in the serum of mice primed with tobacco-derived S1 protein. Enhanced expression of the SARS-CoV N protein was achieved in the tobacco plant *Nicotiana benthamiana* by including the post-transcriptional gene silencing suppressor p19 protein from tomato bushy stunt virus [149]. Intraperitoneal administration of plant extract in BALB/c mice elicited N protein-specific IgG antibodies and, overall, it could be concluded that plant-based expression of the SARS-CoV N protein can induce strong humoral and cellular immune responses in mice. Finally, recent progress in plant-based expression systems has allowed the production of numerous antigens and monoclonal antibodies in plants [158], which has now also accelerated the application of plant-based expression systems for rapid vaccine development against SARS-CoV-2.

Viral vectors have been frequently used as delivery vehicles for immunization against infectious agents such as pathogenic viruses [159]. In many cases, strong humoral and cellular immune responses have been observed as well as protection against challenges with lethal doses of pathogenic viruses. In the context of coronaviruses, a recombinant adenovirus vector expressing the heat-labile enterotoxin B (LTB) and the core neutralizing epitope (COE) of PEDV was administered intramuscularly or orally into BALB/c mice and piglets [150]. Three vaccinations at two-week intervals generated robust humoral and cellular immune responses. Cell-mediated immune responses were seen in mice and neutralizing antibodies inhibited both the vaccine strain and emerging PEDV isolates. Strong immune responses were observed in piglets, but further studies are required to verify the protection against challenges with highly virulent PEDV strains. In another study, the modified vaccinia virus Ankara (MVA) strain was utilized for the expression of recombinant MERS-CoV S protein [151]. Immunization of dromedary camels elicited mucosal immunity. Immunized camels showed significantly reduced excretion of infectious virus and viral RNA transcripts and protection against MERS-CoV correlated with the presence of neutralizing antibodies in the serum. In another application of MVA, the MERS-CoV N was used for induction of cellular immune responses [152]. Identification of MHC-I- and MHC-II-restricted T cell responses was carried out on overlapping peptides spanning the whole MERS-CoV N polypeptide in BALB/c mice immunized with MVA-MERS-CoV N. An H2-d restricted decamer peptide epitope showing CD8^+^ T cell antigenicity was identified, which will be further subjected to protection studies in mouse models for MERS-CoV. In the context of SARS-CoV, the N and S protein genes were cloned between the glycoprotein G and polymerase L genes in a highly attenuated safe rabies virus (RV) vector [153]. A single injection with the RV-SARS-CoV S vector induced strong neutralizing antibody responses in mice, which makes it a promising candidate for eradication of SARS-CoV from animal reservoirs.

Finally, self-replicating RNA viruses have proven to be efficient as vaccine vectors due to the highly efficient replication of their ssRNA genome of positive polarity directly in the cytoplasm [159]. As vectors have been engineered allowing immunization of plasmid DNA replicons, naked RNA replicons or replication-deficient particles, the application range is wide. Moreover, due to self-replication, the amounts required for obtaining similar immune responses as seen for conventional DNA plasmids or synthetic mRNA molecules are 100- to 1000-fold lower [160]. In the context of coronaviruses, Venezuelan equine encephalitis virus (VEE) replicon particles were used for the expression of the Urbani SARS-CoV S or N proteins [154]. It was demonstrated that VEE particles expressing SARS-CoV S but not N provided complete short- and long-term protection against challenges with homologous strains in both young and senescent mice. To evaluate challenges of heterologous strains, a chimeric virus (icGDO3-S) encoding a synthetic S gene of the most genetically divergent human GDO3 strain was engineered. The chimeric virus was highly resistant to neutralization with antisera directed against the Urbani strain. However, immunization with VEE particles expressing SARS-CoV S provided complete short-term protection against challenges with icGDO3-S in young mice, but not in senescent mice. The failure to protect senescent mice was addressed in another study, where mice were vaccinated with VEE particles expressing SARS-CoV S antigen packaged with either attenuated (3014) or wild-type (3000) VEE glycoproteins [155]. The results revealed that aged animals immunized with VEE (3000)-based vaccine were protected against SARS-CoV, while mice immunized with VEE (3014)-based vaccine were not. Furthermore, the superior protection was also confirmed for challenges with influenza virus.

Related to the safety of vaccine development, antibody-dependent enhancement (ADE) comprises an important mechanism facilitating viral cell entry, in which virions are complexed with antibodies, resulting in enhanced viral replication [161]. In the context of COVID-19, prior infection with other coronaviruses, from common cold to SARS-CoV, may have primed COVID-19 patients, causing severe disease once infected with SARS-CoV-2. This could explain the discrepancy between the severity of disease in COVID-19 patients in the Hubei province in China compared to other regions of the world [162]. Recently, a novel molecular mechanism for ADE was revealed, demonstrating that a neutralizing antibody binding to the coronavirus S protein can trigger a conformational change of the S protein mediating viral entry via IgG Fc cellular receptors [163]. Another issue related to vaccine development comprises the type II cellular immunity for lung pathogenicity. In this context, accumulated evidence suggests that the lungs are a major site of immune regulation [164]. Therefore, a highly regulated immune response in the lungs can protect from pathogen infection. In contrast, inefficient immune responses can trigger various pulmonary diseases. Related to SARS and COVID-19 lung pathology, recovery requires rigorous innate and acquired immune responses and epithelial regeneration [165]. However, administration of epithelial growth factors such as the keratinocyte growth factor (KGF) might stimulate the production of ACE2-expressing cells, increasing the viral load.

An immunotherapeutic approach of great interest relates to the use of plasma from convalescent COVID-19 patients. Convalescent plasma has previously been used successfully as post-exposure prophylaxis and/or treatment of SARS and MERS [166]. In the context of COVID-19, in a case study, a patient with severe COVID-19 was treated with convalescent plasma from six donors [167]. The anti-SARS-CoV IgM responses from the convalescent plasma were weak, but high titers of IgG were obtained. The treatment allowed the patient to be released from mechanical ventilation 11 days after plasma transfusion and then transferred to a general ward. In another case study, five critically ill COVID-19 patients with acute respiratory distress syndrome (ARDS) were treated with convalescent plasma obtained from patients who had recovered from COVID-19 [168]. The patients who were receiving mechanical ventilation and had been treated with antiviral agents and methylprednisolone showed a normalized body temperature within 3–4 days and their viral load decreased and became negative within 12 days. Moreover, SARS-CoV-2-specific ELISA and neutralizing antibody titers increased, ARDS resolved in four patients after 12 days and three patients did not need mechanical ventilation after two weeks. Three patients were discharged from the hospital and the two remaining patients were in stable condition. Obviously, the limited number of patients and study design do not permit an evaluation of the efficacy of the treatment and further larger clinical trials are required.

## 4. Conclusions and Future Aspects

In summary, there are no efficient antiviral drugs or vaccines currently available for COVID-19. However, due to the extent of the current COVID-19 pandemic, it seems like the whole world has come together to conquer the outbreak. There seems to be enormous political and economic will to support research and development efforts in an unprecedented way. The positive outcome is that all avenues are explored including antiviral drugs in the form of existing drugs for other viral diseases, which should be subjected to thorough well-planned clinical evaluation. Moreover, new drugs targeting SARS-CoV receptors in the form of small molecules and monoclonal antibodies and gene silencing approaches preventing SARS-CoV replication are being explored.

As drug development and particularly antiviral drugs used for other indications have seen a renaissance of clinical trials for COVID-19, it is appropriate to summarize recent clinical findings. In this context, remdesivir was applied for compassionate use in 53 hospitalized COVID-19 patients—of which, 22 were in the US, 22 in Europe or Canada and 9 in Japan [169]. Thirty-six of the 53 patients (68%) showed clinical improvement, 25 patients were discharged and 7 died. Additionally, COVID-19 patients were subjected to a clinical trial for remdesivir in China [170]. Of the 237 patients enrolled in the study, 158 received remdesivir and 79 were in the placebo group. The study indicated that Remdesivir was not associated with statistically significant clinical benefits although numerical reduction in time to clinical improvement was observed. On 29 April 2020, Gilead, the manufacturer of remdesivir, put out a press release on preliminary data on their phase III SIMPLE trials in COVID-19 patients [171]. The first of the two randomized, open-label, multicenter SIMPLE trials in 397 patients demonstrated that the time of clinical improvement for 50% of patients was 10 days in the group receiving intravenous remdesivir for 5 days and 11 days after 10 days of treatment compoared to 15 days for the control group and more than half of the patients in both groups were discharged from the hospital by day 14. The plan is to extend the study by enrolling an additional 5600 patients and conducting trials at 180 sites in China, France, Germany, Hong Kong, Italy, Japan, Korea, the Netherlands Singapore, Spain, Sweden, Switzerland, Taiwan, the United Kingdom and the United States. In the second SIMPLE trial, the safety and efficacy of 5- and 10-day dosing duration of intravenous remdesivir will be compared to standard of care treatment of COVID-19 patients with the first results expected by the end of May 2020. 

Regarding hydroxychloroquine, initial clinical evaluation in 20 patients was conducted in France, as described earlier [106]. Although the study indicated some therapeutic effect of hydroxychloroquine and at least 80 trials on chloroquine, hydroxychloroquine, or their combination with other drugs have been registered globally, one should be cautious, as many proposals are based on in vitro studies, animal models or experiences from other viral diseases [172]. For instance, a Chinese trial in more than 100 patients demonstrated superiority of chloroquine phosphate compared to control treatment in inhibition of pneumonia exacerbation [173]. Moreover, in a placebo-controlled randomized trial of two different doses of hydroxychloroquine in 62 patients with radiological conformation of pneumonia, but without severe hypoxia, small improvements in body temperature and cough were registered only for the higher dose [174]. However, the results from the lower dose were not described, the endpoints specified in the published protocol differed from those reported and the trial was terminated prematurely [175]. Supporters of hydroxychloroquine have referred to the history of wide and safe use of the drug. However, hydroxychloroquine will potentially expose patients to serious cutaneous adverse reactions [176], hepatic failure [177] and ventricular arrhythmias when co-administered with azithromycin [178].

In the context of lopinavir/ritonavir, a randomized, controlled, open-label trial conducted on 199 COVID-19 patients showed no difference related to clinical improvement compared to the standard-care control group [87]. Detectable viral RNA levels and mortality numbers were similar for the two groups. Gastrointestinal adverse events were more common for patients treated with lopinavir/ritonavir, but serious adverse events were more frequent in the standard-care group. The Efficacy of Lopinavir Plus Ritonavir and Arbidol against Coronavirus Infection (ELACOI) single-blind randomized controlled trial in China included 44 patients with mild or moderate clinical status taking lopinavir/ritonavir or arbidol (Umifenovir) [179]. The study showed no differences in the time to negative pharyngeal SARS-CoV-2 PCR detection, pyrexia, cough or lung CT findings between the treatment and control groups. However, in the lopinavir/ritonavir group, 38.1% of patients deteriorated to severe/critical status compared to 12.5% and 14.3% for the arbidol and control groups, respectively. No adverse events were registered in the arbidol and control groups. In contrast, gastrointestinal and deranged liver adverse events were seen after lopinavir/ritonavir treatment.

Overall, although drugs developed for other infectious diseases might provide alternative treatment strategies for COVID-19 when evaluated in well-designed and well-executed clinical trials, development of prophylactic drugs and vaccines targeting specific structures of SARS-CoV-2 seems like the preferred approach. In the context of vaccine development, utilization of purified epitope peptides and antigens and cellular or viral delivery systems should all be explored in parallel to achieve maximum success in a minimum time frame. Not surprisingly, more than 60 vaccine projects are in progress at pre-clinical or clinical levels [180]. These initiatives apply all possible means of delivery including mRNA, plasmid DNA, non-replicating viral vectors, inactivated and live attenuated virus and protein subunits.

It will also be necessary to look ahead to the possibility of preventing SARS-CoV-2 outbreaks in the future and, if not, to be better prepared for a second or third wave of coronavirus or any other pandemic. It is important to understand the origin of SARS-CoV-2 rather than place blame on bats for the cause of the pandemic [181]. On several occasions, it has been pointed out that bats are the only flying mammals. How wrong is this? In fact, human beings are the real flying mammals, jetting around the world at an ever-accelerating pace and frequency. Only last year, an estimated 4.5 billion passengers took to the skies. Furthermore, in defense of bats, recent findings suggest that pangolins are the prime suspects as the source of SARS-CoV-2 although it has yet to be confirmed [182]. In any case, all trails lead back to humans, as pangolins are sought after for their meat and scales. Instead of the blame game, the focus should now be on accelerated efforts to develop novel safe and efficacious prophylactic and therapeutic approaches in the form of coronavirus drugs and vaccines.

## Figures and Tables

**Figure 1 biomedicines-08-00109-f001:**
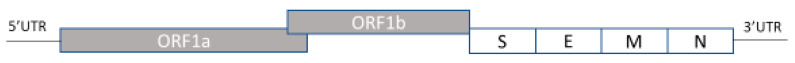
Schematic illustration of the severe acute respiratory syndrome-coronavirus (SARS-CoV)-2 genome. ORF1a and ORF1b encode the non-structural proteins. The structural proteins are encoded by spike (S), envelope (E), membrane (M) and nucleocapsid (N) genes.

**Figure 2 biomedicines-08-00109-f002:**
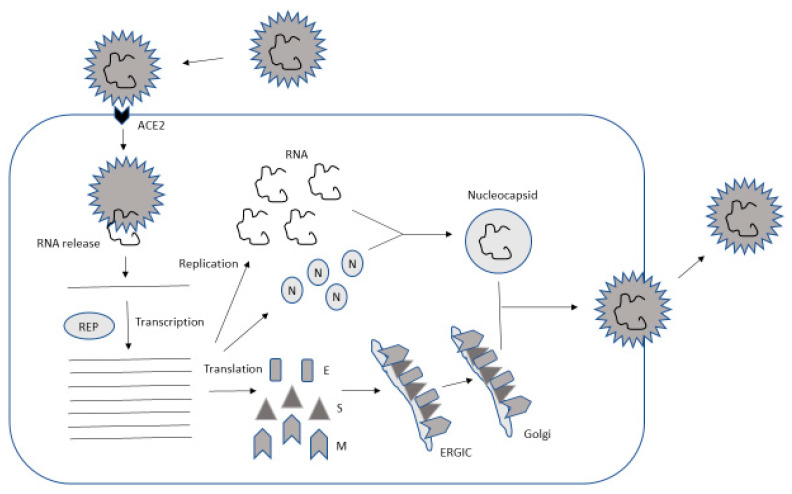
Lifecycle of SARS-CoV-2. The attachment occurs at ACE2, followed by release of viral RNA into the cytoplasm. The replicase (REP) complex is responsible for RNA replication. RNA and the translated nucleocapsid (N) protein form nucleocapsids, while the spike (S), envelope (E) and membrane (M) proteins go through the ER–Golgi intermediate compartment (ERGIC) and Golgi before the assembly of virus particles takes place on the plasma membrane from where mature virions are released.

**Table 1 biomedicines-08-00109-t001:** Coronavirus-based diseases in animals and humans.

Virus	Disease	Effect	Ref.
TGEV	Gastroenteritis in pigs	High morbidity, mortality	[3]
PEDV	Gastroenteritis in pigs	High morbidity, mortality	[4]
PHEV	Enteric infection	Diarrhea, encephalitis	[5]
BCoV	Respiratory tract infection	Significant loss in cattle industry	[6,7]
RCoV	Respiratory tract infection	Model system	[8]
IBV	Respiratory infections, renal disease	Significant losses in the chicken industry	[7,9]
FCoV	Respiratory tract infection	Mild or asymptomatic	[10]
FIPV	Infectious peritonitis	Lethal FIP	[11]
SW1	Respiratory, acute liver	SW1 found in deceased whale	[13]
Bat CoV	Respiratory tract infection	Potential threat of epidemics	[14]
SHC014-CoV	Risk of epidemic	Chimeric CoV presents risk	[15]
MHV	Respiratory, enteric, neurological infections	Mouse model for human disease	[16]
A59, JHMV	Chronic demyelination	Mouse model for MS	[17]
α HCoV-229E	Respiratory infection	15%–30% of annual common cold	[18]
α HCoV-NL63	Respiratory infection	Also associated with croup	[18,19]
β HcoV-OC43	Respiratory infection	15%–30% of annual common cold	[20]
β HcoV-HKU1	Respiratory infection	15%–30% of annual common cold	[20]
SARS-CoV	SARS	Epidemic with 774 deaths	[21,22]
MERS-CoV	MERS	Epidemic with 333 deaths	[23]

α, α-coronavirus; A59, type of MHV; β, β-coronavirus; BCoV, bovine CoV; CoV, coronavirus; FCoV, Feline coronavirus; FIP, feline infectious peritonitis; FIPV, feline infectious peritonitis virus; IBV, infectious bronchitis virus; JHMV, type of MHV; MERS-CoV, Middle East respiratory syndrome-coronavirus; MHV, mouse hepatitis virus; MS, multiple sclerosis; PEDV, porcine epidemic diarrhea virus; PHEV, porcine hemagglutinating encephalomyelitis virus; RCoV, rat CoV; SARS-CoV, severe acute respiratory syndrome-coronavirus; TGEV, transmissible gastroenteritis virus (TGEV); SW1, whale coronavirus.

**Table 3 biomedicines-08-00109-t003:** Gene silencing against coronaviruses.

Therapy	Disease	Effect	Ref.
siRNAs	SARS-CoV	Inhibition of replication in Vero E6 cells	[119]
	SARS-CoV	Prophylactic/therapeutic effects in FRhK4 cells	[120]
	SARS-CoV	Suppression of SARS symptoms in primates	[121]
	SARS-CoV	Reduced infection in ACE2-silenced cells	[122]
siRNAs, miRNAs	SARS-CoV	Knockdown of ezrin	[123]
	MERS-CoV	Design of siRNAs, miRNAs for MERS control	[124]
siRNAs	FIPV	siRNA combination delays resistance	[125]
shRNAs	PDCoV	Reduced titers and viral RNA in ST cells	[126]
	PEDV	Inhibition of viral RNA and replication	[127]
	SADS-CoV	Inhibition of viral RNA and replication	[127]
	PDCoV	Inhibition of viral RNA and replication	[127]

FIPV, feline infectious peritonitis virus; MERS-CoV, Middle East respiratory syndrome-coronavirus; miRNAs, micro RNAs; PDCoV, porcine deltacoronavirus; PEDV, porcine epidemic diarrhea virus; SADS-CoV, swine acute diarrhea syndrome-coronavirus; SARS-CoV, severe acute respiratory syndrome-coronavirus; shRNAs, short hairpin RNAs; siRNAs, short interfering RNAs; ST, swine testicular.

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
