# Peer review of "Coronavirus Pandemic—Therapy and Vaccines"

_biomedicines, 2020, doi:10.3390/biomedicines8050109_

Round 1

Reviewer 1 Report

This is a very extensive review and a well-written article for coronavirus background and the development of therapeutics and vaccines against the coronavirus pandemics. The author has covered most of the animal and human coronaviruses and summarized in an retrospective way. The vaccine development part is also well-written with the extensive review on different vaccine platform technology. Only the vaccine safety issue for coronavirus-induced antibody dependent enhancement  and type II T cellular immunity for lung pathogenicity should be included. Basically, this is an excellent review article to meet publication standards.

Author Response

Dear Reviewer 1,

Text has been added on "coronavirus induced antibody dependent enhancement and type II T cellular immunity in relation to lung pathogenicity" 

Reviewer 2 Report

  • General:

Very informative and comprehensive manuscript. The introduction and virology sections set important context for the discussion regarding therapeutics and vaccines. 

  • Abstract:

"spread of COVID-19"

Consider, spread of SARS-CoV-2...

"which will be useful"

Consider, may be useful.

  • Introduction

page 4, line 2 - flu season

Consider, return in a seasonal or cyclical pattern...

  • Therapeutic agents

This section of the manuscript would benefit from a short update on the clinical experience with some of the agents described. Remdesivir, hydroxychloroquine, and lop/rit have all posted clinical data. The manuscript would be strengthened by including. 

The paper would also be strengthened by adding a section of convalescent plasma. This approach not only has importance as a stand alone therapy with potential for clinical benefit but it speaks to a potential mechanism of action would could be leveraged to inform monoclonal and vaccine strategies. 

  • Vaccine development

A broader review of the vaccine constructs under active clinical development would strengthen the manuscript. A review of clinicaltrials.org would be valuable. 

Author Response

Dear Reviewer 2,

Thank you for your much appreciated comments/suggestions. Please find below my responses.

"spread of COVID-19" changed to "spread of SARS-CoV-2"

"flu season" changed to "return in a seasonal pattern"

A paragraph on "the latest of Remdesivir, hydroxychloroquine, and lop/rit drug trials" has been added

A paragraph on "convalescent plasma" has been added

Text has been added to "vaccine constructs" and a reference included for ongoing preclinical and clinical studies